# Environmental Risk Factors and Health: An Umbrella Review of Meta-Analyses

**DOI:** 10.3390/ijerph18020704

**Published:** 2021-01-15

**Authors:** David Rojas-Rueda, Emily Morales-Zamora, Wael Abdullah Alsufyani, Christopher H. Herbst, Salem M. AlBalawi, Reem Alsukait, Mashael Alomran

**Affiliations:** 1Department of Environmental and Radiological Health Sciences, Colorado State University, Environmental Health Building, 1601 Campus Delivery, Fort Collins, CO 80523, USA; 2Logan Simpson, 213 Linden Street, Fort Collins, CO 80524, USA; moralesemily@yahoo.com; 3Saudi Center for Disease Prevention and Control, 70 SCDC Building, Al Aarid, King Abdulaziz Rd, Riyadh 13354, Saudi Arabia; Alsufyaniw@moh.gov.sa (W.A.A.); albalawis@moh.gov.sa (S.M.A.); msalomran@moh.gov.sa (M.A.); 4Health, Nutrition and Population Global Practice, The World Bank, Diplomatic Quarter, Riyadh Country Office, Riyadh 94623, Saudi Arabia; cherbst@worldbank.org (C.H.H.); ralsukait@worldbank.org (R.A.); 5Community Health Department, King Saud University, Riyadh 11433, Saudi Arabia

**Keywords:** environmental risk factors, umbrella review, meta-analyses, systematic review, epidemiological studies

## Abstract

**Background**: Environmental health is a growing area of knowledge, continually increasing and updating the body of evidence linking the environment to human health. **Aim**: This study summarizes the epidemiological evidence on environmental risk factors from meta-analyses through an umbrella review. **Methods**: An umbrella review was conducted on meta-analyses of cohort, case-control, case-crossover, and time-series studies that evaluated the associations between environmental risk factors and health outcomes defined as incidence, prevalence, and mortality. The specific search strategy was designed in PubMed using free text and Medical Subject Headings (MeSH) terms related to risk factors, environment, health outcomes, observational studies, and meta-analysis. The search was limited to English, Spanish, and French published articles and studies on humans. The search was conducted on September 20, 2020. Risk factors were defined as any attribute, characteristic, or exposure of an individual that increases the likelihood of developing a disease or death. The environment was defined as the external elements and conditions that surround, influence, and affect a human organism or population’s life and development. The environment definition included the physical environment such as nature, built environment, or pollution, but not the social environment. We excluded occupational exposures, microorganisms, water, sanitation and hygiene (WASH), behavioral risk factors, and no-natural disasters. **Results**: This umbrella review found 197 associations among 69 environmental exposures and 83 diseases and death causes reported in 103 publications. The environmental factors found in this review were air pollution, environmental tobacco smoke, heavy metals, chemicals, ambient temperature, noise, radiation, and urban residential surroundings. Among these, we identified 65 environmental exposures defined as risk factors and 4 environmental protective factors. In terms of study design, 57 included cohort and/or case-control studies, and 46 included time-series and/or case-crossover studies. In terms of the study population, 21 included children, and the rest included adult population and both sexes. In this review, the largest body of evidence was found in air pollution (91 associations among 14 air pollution definitions and 34 diseases and mortality diagnoses), followed by environmental tobacco smoke with 24 associations. Chemicals (including pesticides) were the third larger group of environmental exposures found among the meta-analyses included, with 19 associations. **Conclusion**: Environmental exposures are an important health determinant. This review provides an overview of an evolving research area and should be used as a complementary tool to understand the connections between the environment and human health. The evidence presented by this review should help to design public health interventions and the implementation of health in all policies approach aiming to improve populational health.

## 1. Introduction

In 2012, the World Health Organization (WHO) estimated that 12.6 million global deaths, representing 23% (95% CI: 13–34%) of all deaths, were attributable to the environment [1]. Air pollution and second-hand smoke are responsible for 52 million lower-respiratory diseases each year, representing 35% of the global cases [1]. Non-communicable diseases are also related to air pollution, chemicals, and second-hand smoke, which are responsible for 119 million cardiovascular diseases each year, 49 million cancers, and 32 million chronic respiratory diseases [1]. Environmental risks to health include pollution, radiation, noise, land use patterns, or climate change [2]. 

Environmental health is a growing area of knowledge, continually increasing and updating the body of evidence linking the environment to human health. The Global Burden of Disease project considers 26 environmental and occupational risk factors in their estimations [3]. Such risk factors are those that have enough evidence to be translated with available global exposure data to quantify their impact across the globe. However, these are far from representing the totality of evidence related to environmental exposures and human health.

Global populations are also facing population growth and aging, increasing groups vulnerable to environmental risk factors. Around 10% of the global gross domestic product is spent on healthcare [2], but little is allocated to primary prevention and public health. Be able to identify environmental risk factors is crucial in the decision-making process aiming to protect public health. The investment in measures and policies aiming to reduce environmental risks could help alleviate the health burden that health care systems around the globe are facing.

This study aims to provide an overview of the most recent evidence linking environmental risk factors and health outcomes. Applying an umbrella review approach, this study presents a synthesis of the epidemiological evidence from meta-analyses. The umbrella review systematically identifies and selects the available scientific publications in a research area. The review focuses on meta-analyses from cohort, case-control, case-crossover, and time-series observational studies, relating short and long-term environmental exposures to morbidity and mortality. The review summarizes the statistically significant associations reported in the latest published meta-analysis with the largest available number of individual studies and populations. 

## 2. Methodology

This study is a systematic collection and assessment of multiple systematic reviews with meta-analyses performed on a specific research topic, also known as an umbrella review. The methods of the umbrella review are standardized. In this work, we follow state-of-the-art approaches, as in previously published umbrella reviews on risk factors for health outcomes [4]. The study protocol was developed in accordance with the reporting guidance in the Preferred Reporting Items for Systematic Reviews and Meta-Analyses Protocols (PRISMA-P) statement and registered in the International Prospective Register of Systematic Reviews (PROSPERO—CRD42020196152).

### 2.1. Literature Search

A search strategy was designed to identify studies published in Medline via PubMed. The search strategy identified systematic reviews of observational studies with a meta-analysis that evaluated the associations between environmental risk factors and health outcomes defined as incidence, prevalence, and mortality. We further hand-searched reference lists of the retrieved eligible publications to identify additional relevant studies. The specific search strategy included free text and Medical Subject Headings (MeSH) terms related to risk factors, environment, health outcome, observational studies, and meta-analysis. The search was limited to English, Spanish, and French published articles and studies on humans. The last search was conducted on 20 September 2020. The results of the searches were cross-checked to eliminate duplicates.

#### Search Strategy

(“Risk Factors” [Mesh]) OR risk factor OR Environmental risk factors)

AND

(Environment * OR “Environment”[Mesh] OR Environmental pollution OR “Environmental Pollution”[Mesh] OR Environmental exposures OR “Environmental Exposure”[Mesh] OR Environment Design OR “Environment Design”[Mesh] OR Built Environment OR “Built Environment”[Mesh] OR Environmental Medicine OR “Environmental Medicine”[Mesh] OR Decontamination OR “Decontamination”[Mesh])

AND

(Health OR “Health”[Mesh] OR Health Outcome OR Population Health OR “Population Health”[Mesh] OR Pathological Conditions OR “Pathological Conditions, Signs and Symptoms”[Mesh] OR Pathologic Processes OR “Pathologic Processes”[Mesh] OR Disease OR “Disease”[Mesh] OR Syndrome OR “Syndrome”[Mesh] OR Morbidity OR “Morbidity”[Mesh] OR Incidence OR “Incidence”[Mesh] OR Prevalence OR “Prevalence”[Mesh] OR Mortality OR “Mortality”[Mesh] OR Death OR “Death”[Mesh] OR Cause of Death OR “Cause of Death”[Mesh] OR Life Expectancy OR “Life Expectancy”[Mesh])

AND

(Longitudinal Studies OR “Longitudinal Studies”[Mesh] OR Observational Study OR “Observational Study” [Publication Type] OR Cohort Studies OR “Cohort Studies”[Mesh] OR Case-Control Studies OR “Case-Control Studies”[Mesh] OR Time Series OR “Interrupted Time Series Analysis”[Mesh])

AND

(Meta-Analysis OR “Meta-Analysis” [Publication Type])

NOT

(“Social Environment” [MeSH Terms] OR Social Environment)

### 2.2. Selection Criteria

We included meta-analyses of cohort, case-control, case-crossover, and time-series studies examining associations between health outcomes and potential environmental risk factors. Health outcomes were defined as disease incidence, prevalence, cause-specific mortality, and all-cause mortality. Risk factors were defined as any attribute, characteristic, or exposure of an individual that increases the likelihood of developing a disease or death. The environment was defined as the external elements and conditions that surround, influence, and affect a human organism or population’s life and development. The environment definition included the physical environment such as nature, built environment, or pollution, but not the social environment. We excluded occupational exposures, microorganisms, water, sanitation and hygiene (WASH), behavioral risk factors, and no-natural disasters. We only included meta-analyses that reported statistically significant pooled effect estimates and confidence intervals (CI) from observational studies. When two or more meta-analyses existed for an association, we included the most recent meta-analysis with the largest number of studies and populations. We chose eligible articles by consecutively examining the titles, abstracts, and the full-text. Two investigators (DRR and EMZ) independently and blindly screened the titles and abstracts to determine the articles’ inclusion. Eligibility criteria were applied to the full-text articles during the final selection. We manually searched the references of the relevant articles and attempted to identify and include eligible studies. Disagreements were resolved via discussion between reviewers.

### 2.3. Data Extraction and Analysis

Data extracted from each meta-analysis included the first author, publication year, environmental risk factor, exposure unit or exposure comparator, exposure temporality, study design, population, health outcome, number of studies included, summary meta-analytic estimates (i.e., odds ratio or relative risk) and corresponding 95% CI, random effect p-value, and heterogeneity measure. A narrative synthesis of the included meta-analyses was carried out by environmental risk factors, health outcomes, and population. 

To assess the strength of epidemiologic evidence, we considered the estimate’s precision and the results’ consistency. We noted which associations met the following criteria: (1) precision of the estimate (i.e., *p* < 0.001, a threshold associated with significantly fewer false-positive results), and (2) consistency of results (I2 < 50%). The strength of the epidemiologic evidence was rated as high (when both criteria were satisfied), moderate (if 1 consistency of results was not satisfied), or low (if both consistencies of results were not satisfied.

## 3. Results 

### 3.1. Literature Review

We identified 1266 publications in PubMed and 87 publications through a hand search (Figure 1). We excluded 1137 (89%) publications after screening the titles and abstracts for duplications or for not meeting our inclusion criteria. After, we reviewed the full texts of the remaining 216 (11%) publications. From these publications, 1 publication was excluded because it did not report a meta-analysis, 7 because they did not include an environmental risk factor, 10 due to the lack of statistical significance in the pooled meta-estimates, 17 because the meta-analysis did not include cohort, case-control, case-crossover, and time-series studies or combined cross-sectional studies with cohort or case-control studies, 26 because they did not report morbidity (incidence or prevalence) or related mortality estimates, and finally, 52 others because the studies they did not provide the latest available evidence and/or the largest sample size. 

In total, 103 publications associating environmental risk factors and health outcomes through were included in this umbrella review. These studies include a total of 69 environmental risk factors that were grouped in air pollutants (14 risk factors), environmental tobacco smoke (6 risk factors), chemicals and heavy metals (25 risk factors), physical exposures (14 risk factors), and surrounding residential exposures (10 risk factors). On average, the meta-analysis included 37 studies ranging from 2 to 652. In terms of study design, 57 included cohort and/or case-control studies, and 46 included time-series and/or case-crossover studies. In terms of the study population, 1 included the elderly, 1 included only men, 13 included only women, 21 included children, and the rest included adult population and both sexes. From all the meta-analyses included, 9 were published before 2013, 13 were published in 2014, 7 in 2015, 11 in 2016, 13 in 2017, 14 in 2018, 24 in 2019, and 12 in 2020. In total, the studies reported 72 different long- and short-term diseases or mortality diagnoses.

### 3.2. Air Pollution

We identified 14 air pollutants related to 34 diseases and mortality diagnoses. The air pollutant with the most extensive list of health impacts (29 diagnoses) was the particulate matter with less than 2.5 micrometers of diameter (PM2.5), followed by particulate matter with less than 10 micrometers of diameter (PM10) (17), nitrogen dioxide (NO2) (17), ozone (O3) (7), household air pollution (5), sulfur dioxide (SO2) (4), carbon monoxide (CO) (4), solid fuel use (4), nitrogen oxides (2), desert dust (2), biomass burning (2), black carbon (1), and indoor air pollution from solid fuel (1). Air pollution was reported to affect all age groups and both sexes. 

Long-term impacts of particulate matter (PM2.5 and PM10) were reported for 35 diagnoses and causes of death (Table 1, Table 2 and Table 3). Adults exposed to PM2.5 or PM10 reported an increased risk of chronic kidney disease [5], type 2 diabetes [6], lung cancer mortality [7,8], and cancer mortality [7]. Adults exposed to PM2.5 also reported an increased risk of Alzheimer’s disease [9], all-cause mortality [10], cardiovascular mortality [11], chronic obstructive pulmonary disease (COPD) [8], colorectal cancer mortality [7], dementia [9], depression [12], ischemic heart disease (IHD) mortality [8], liver cancer mortality [7], natural mortality [11], respiratory mortality [11], stroke [13], stroke mortality [8] and Parkinson’s disease [14]. Adults exposed to PM10 reported an increased incidence of coronary events [15] and chronic bronchitis [16]. Pregnant women exposed to PM2.5 reported an association with offspring diagnosis of autistic syndrome disorder [17], small for gestational age [18], and those exposed to PM10 reported an association with low birth weight [18] and preterm birth [18]. For children, exposure to PM2.5 was associated with asthma [19], acute respiratory infections [8], and autistic spectrum disorder [20]. Moreover, children’s exposure to PM10 was also associated with an increased risk of asthma [19] and autistic spectrum disorder [20].

Particulate matter that includes PM2.5 and PM10 reported six diagnoses and causes of death related to short-term exposures (Table 2)**.** In adults, short-term exposure to PM2.5 and PM10 were associated with out-of-hospital cardiac arrest [21], cardiac arrhythmia [22], daily cardiovascular, respiratory, and natural mortality [23]. In addition, for PM10, suicide was also reported as a short-term impact [12]. In children, short-term exposure to PM2.5 or PM10 was associated with pneumonia [24].

Desert dust, an important natural source of particulate matter, was also associated with health impacts (Table 3)**.** This review identified one meta-analysis of adult exposure to desert dust, reporting an increased risk of cardiovascular mortality and natural mortality [25]. Another component of particulate matter is black carbon, which originates from fossil fuel and biomass combustion. We identified one meta-analysis on black carbon in children reporting an increased risk of asthma [19]. 

Nitrogen oxides (NOx and NO2) were associated with 18 different diagnoses and causes of death (Table 4)**.** Pregnant women’s exposure to NOx was associated with low birth weight [18] and preterm birth [18]. For the same group, exposure to NO2 reported an increased risk of low birth weight [18] and small for gestational age [18]. For adults, long-term exposure to NO2 was associated with an increased risk of all-cause mortality [11], autistic syndrome disorder [17], cancer mortality [7], cardiovascular mortality [11], chronic kidney disease [5], cancer mortality [7], respiratory mortality [11], and type 2 diabetes [26]. Furthermore, for adults, short-term exposure to NO2 was associated with an increased risk of out-of-hospital cardiac arrest [21], cardiac arrhythmia [22], conjunctivitis [27], depression [28], and natural mortality [16]. Lastly, children’s long-term exposure to NO2 was associated with an increased risk of asthma [19], and short-term exposure with an increased risk of pneumonia [24].

Ozone (O3) was found as a risk factor for seven diagnoses and causes of death (Table 5). Long-term exposure to O3 was reported to increase IHD mortality [29] and Parkinson’s disease [14] in adults and for pregnant women with preterm birth [18]. Short-term exposure to ozone was associated as a risk factor for pneumonia in children [24] and in adults with out-of-hospital cardiac arrest [21], all-cause mortality [16], and cardiovascular and respiratory mortality [16]. 

Sulfur dioxide (SO2) is a prevalent pollutant and was found as a risk factor for four diagnoses (Table 5). SO2 is a gas primarily emitted from fossil fuel combustion at power plants and other industrial facilities as well as from fuel combustion in mobile sources like locomotives or ships. In their first trimester, pregnant women exposed to SO2 reported an increased risk of gestational diabetes mellitus [30]. Pregnant women exposed during any trimester also reported an increased risk of low birth weight [18]. Short-term exposures to SO2 were associated with pneumonia in children [24] and cardiac arrhythmia in adults [22].

Carbon monoxide (CO) is a gas produced by fuel combustion in motorizing vehicles, small engines, stoves, and fireplaces, among others (Table 5). We identified four health impacts associate with CO exposure. In short term exposures, CO was reported as a risk factor for pneumonia in children [24], and cardiac arrhythmia [22], and out-of-hospital cardiac arrest in adults [21]. CO exposure during pregnancy was also reported as a risk factor for preterm birth [18].

Household air pollution represents indoor air pollution from multiple sources (e.g., cooking and heating) (Table 6). Under this review, we identified five types of cancers related to household air pollution exposure. Specifically, one meta-analysis reported an increased risk for cervical, laryngeal, nasopharyngeal, oral, and pharyngeal cancers [31]. Indoor air pollution from solid fuels was also found as a risk factor for hypertension [32]. Solid fuel use by pregnant women was associated with low birth weight, stillbirth, preterm birth, and intrauterine growth retardation in another meta-analysis [33]. Finally, biomass burning was associated with an increased risk of esophageal squamous cell carcinoma [34] and COPD [35].

### 3.3. Environmental Tobacco Smoke

Environmental tobacco smoke is an involuntary exposure to tobacco smoke, also known as passive smoke or secondhand smoke. Environmental tobacco smoke is generated by tobacco products’ combustion and is a complex mixture of over 4000 compounds. These include more than 40 known or suspected human carcinogens, such as 4-aminobiphenyl, 2-naphthylamine, benzene, nickel, and various polycyclic aromatic hydrocarbons (PAHs) and N-nitrosamines. Furthermore present are several irritants, such as ammonia, nitrogen oxides, sulfur dioxide, and aldehydes, and cardiovascular toxicants, such as carbon monoxide, nicotine, and some PAHs [37,38]. 

This review identified 23 diseases and causes of death related to environmental tobacco smoke, parental, and prenatal smoke (Table 7). Specifically, environmental tobacco smoke was reported to be associated in adults with stroke [39], lung cancer in women [40], and in pregnant women with low birth weight [37] and small for gestational age [37]. Passive smoking was associated in adults with an increased risk of breast cancer [41], cardiovascular disease [42], cervical cancer [43], lung cancer, lung adenocarcinoma, large cell lung cancer, small cell lung cancer, squamous cell carcinoma [44], all-cause mortality [42], and type 2 diabetes [45]. In pregnant women, passive smoking was associated with neural tube defects [46]. In children, passive smoking was associated as a risk factor for asthma [47] and otitis media [48]. Prenatal smoke was found to be associated with schizophrenia [49], offspring depression [50], and attention-deficit/hyperactivity disorder [51]. Parental smoke with childhood obesity [52], maternal smoke with neuroblastoma [53], and paternal smoke with acute myeloid leukemia [54] and acute lymphoblastic leukemia [55].

### 3.4. Chemicals, Pesticides, and Heavy Metals

This review identified two health outcomes associated with childhood exposure to 1,3-butadiene (Table 8). 1,3-Butadiene is a synthetic gas used primarily as a monomer to manufacture many different polymers and copolymers and as a chemical intermediate in industrial chemical production. Motor vehicle exhaust is also a source of 1,3-butadiene. One meta-analysis found that long-term exposure to 1,3-Butadiene during childhood increased the risk of acute lymphoblastic leukemia and all leukemias [56]. Another group of chemicals found to be associated with health impacts were the hydrocarbons (Table 8). Hydrocarbons are present in a broad range of products, including petroleum and other fuels, solvents, paints, glues, and cleaning products [57]. A meta-analysis of 14 studies showed that long-term exposure to hydrocarbons was associated with Parkinson’s disease [58]. Organic solvents and other solvents were also found to be associated with neurological and rheumatological diseases (Table 8)**.** Specifically, long-term exposure to organic solvents was associated with multiple sclerosis [59] and Parkinson’s disease [58]. Long-term exposure to solvents was also found to be associated with an increased risk of systemic sclerosis [60]. Organic solvents are used in many industries. They are used in paints, varnishes, lacquers, adhesives, glues, and degreasing and cleaning agents, and the production of dyes, polymers, plastics, textiles, printing inks, agricultural products, and pharmaceuticals.

In adults, long-term exposure to polychlorinated biphenyls (PCBs) were found to be associated with non-Hodgkin lymphoma [61], in women with endometriosis [62], and in children (<18 months of age), PCB 153 was found to be associated win increase risk of bronchitis [63] (Table 8). Polychlorinated biphenyls are a large group of human-made organic chemicals that, due to their properties like non-flammability, chemical stability, high boiling point, and electrical insulating capacity, are widely used industrial and commercial applications. Bisphenol A (BPA), a chemical used primarily in the production of polycarbonate plastics and epoxy resins, for example, in food and drink packaging, was found to be a risk factor for diabetes [64] and obesity in adults [64] (Table 8). Women’s exposure to mono-(2-ethyl-5-hydroxyhexyl) phthalate (MEHHP) has been found as a risk factor for endometriosis [65] (Table 8). MEHHP is a metabolite of phthalate acid esters (PAEs). MEHHP is often found in the blood and tissues of the general population. Studies have shown that women are more likely to be exposed to PAEs through products such as perfume, cosmetics, and personal care products. The review found evidence of dioxins as a risk factor for endometriosis [62] (Table 8). Dioxins are a group of chemically-related compounds that are persistent environmental pollutants (POPs). Dioxins are unwanted by-products of a wide range of manufacturing processes, including smelting, chlorine bleaching of paper pulp, manufacturing some herbicides and pesticides, and incinerators. 

Pesticide exposure also was found by multiple meta-analyses as a risk factor for several diseases in adults and children (Table 9 and Table 10). In adults, pesticides, in general, were found to be associated with Alzheimer’s disease [66], amyotrophic lateral sclerosis [67], brain tumors [68], myelodysplastic syndromes [69], and Parkinson’s disease [70]. Organochlorine pesticides were associated with endometriosis [62]. Paraquat, a dichloride pesticide, was also found to be related to Parkinson’s disease [71]. Non-Hodgkin lymphoma was also associated with multiple types of pesticides, like organophosphate [72], organochlorine [73], chlordane [73], diazinon [72], hexachlorobenzene [73], hexachlorocyclohexane [73], and dichlorodiphenyldichloroethylene(DDE) pesticides [73]. Finally, children (<18 months of age) reported a higher risk of bronchitis when exposed to DDE [63], and children’s residential exposure to pesticides was reported as a risk factor for acute lymphoblastic leukemia, acute myeloid leukemia, and childhood leukemia [74]. 

In terms of mineral and heavy metals, aluminum, asbestos, cadmium, chromium, arsenic, lead, and silica, were also associated with multiples health outcomes (Table 11). Aluminum was associated with dementia in adults [45]. Non-occupational asbestos was associated with mesothelioma [75]. Cadmium exposure was associated with cancer, especially lung cancer [76]. Chromium exposure was associated with schizophrenia [77]. Inorganic arsenic was associated with type 2 diabetes [78]. Lead exposure to amyotrophic lateral sclerosis [79] and mild mental retardation [80]. Silica exposure with systemic sclerosis [60].

### 3.5. Physical Exposures

Physical exposures refer to environmental factors such as temperature, noise, or radiation. Our review identified 21 meta-analyses covering 14 physical environmental exposures and 27 different diseases or causes of death among children, women, adults, and elderly populations. Ambient temperature and extreme weather were the most common physical environmental risk factor studied among the meta-analysis found in this review (Table 12). Changes in ambient temperature (increases or decreases) were related to short-term health impacts. Particularly in adults, increases in the ambient temperature above the 93rd percentile were found to be a risk factor of suicide [81], those expose to temperatures above 90th percentile or below 10th percentile to diabetes mortality [82], and those under orthopedic procedure during warmer weather periods of the year had an increased risk of postoperative infection [83]. Comparing high versus low temperatures, high temperature increases the risk of low birth weight and stillbirth among pregnant women [84]. Furthermore, changes in diurnal temperature by increases of 10 degrees Celsius were related to increased mortality [85]. Furthermore, heatwaves, defined as a high temperature lasting for several days, were associated with cardiovascular and respiratory mortality in adults [86] and preterm birth [84]. For the elderly populations, heat changes by 1 Celsius degree increment above a threshold were related to acute renal failure, cardiovascular disease mortality, cerebrovascular mortality, diabetes, ischemic heart disease mortality, respiratory disease, and respiratory mortality [87]. In terms of cold temperatures, reductions of 1 Celsius degree during winter times were related to an increased risk of cardiovascular mortality, cerebrovascular mortality, intracerebral hemorrhage, pneumonia, and respiratory mortality [87]. Cold waves were also associated with cardiovascular mortality [88]. For children, reductions of 1 degree Celsius during cold weather were related to an increased risk of asthma(<12 years old) [89]. 

Natural and artificial light exposure was also associated with positive and negative health impacts (Table 13). Outdoor light exposure was found as a protective factor for myopia in children [90]. The main explanation for this effect is the impact of sunlight on eyeball size, neurotransmitters released in the retina, and vitamin D synthesis. In contrast, artificial light exposure at night was associated as a risk factor for women’s breast cancer [91]. The main explanation for the increased risk of breast cancer is the impact of artificial light on reducing sleep duration and melatonin release. Melatonin is suggested as a carcinogenesis inhibitor; thus, low melatonin concentrations could contribute to breast cancer development. Ultraviolet radiation was found to be a protective factor for positive Epstein–Barr Virus Hodgkin lymphoma in adults [92], and recreational sun exposure was associated with non-Hodgkin lymphoma [93]. 

The noise was another environmental risk factor that was found to be associated with non-communicable diseases (Table 13). In particular, noise exposure from any source was found to be a risk factor for diabetes [94], and each increment of 5 decibels of ambient noise was associated with an increased risk of hypertension [95]. In addition, road traffic noise increments were associated with diabetes [94], hypertension in men [96], and ischemic heart disease [97]. 

Radon, a radioactive natural, was found in a recent meta-analysis as a risk factor for lung cancer [100] at indoor radon exposure levels above 100 Bq/m^3^ (Table 13). In another meta-analysis, indoor radon exposure was also associated as a risk factor for childhood leukemia [101]. Finally, long-term exposures to extremely low-frequency electromagnetic fields were also found associated as a risk factor for amyotrophic lateral sclerosis [67] and childhood leukemia [99] (Table 13). Extremely low-frequency (ELF) magnetic fields are alternating fields generated by the distribution and supply of electricity. 

### 3.6. Residential Surroundings

In this category, we summarized the environmental exposures related to residential surroundings, such as greenness, proximity to roadways and petrochemical complexes, or the degree of urbanization. We also located other residential exposures, such as the presence of pets that are suggested as a protective factor for non-communicable diseases. We identified two meta-analyses associating residential greenness as a protective factor for adults and newborns health (Table 14). Specifically, we found evidence that greenness in a 300 m buffer around homes was associated with a reduced risk for mortality in adults [102] and a reduced risk of low birth weight [103]. In addition, residential greenness in a 500 m buffer from homes was also associated with a reduced risk of newborns being small for their gestational age [103]. Living near major roadways or being exposed to traffic around homes was found as a risk factor for type 2 diabetes in adults [104] and leukemia in children [105] (Table 14). Living near petrochemical industrial complexes was also found to produce multiple types of leukemias (Table 14). Specifically, living in an 8 km radius from a petrochemical complex was found to be a risk factor for acute myeloid leukemia, chronic lymphocytic leukemia, and all leukemias [106].

The degree of urbanization was also related to several health impacts (Table 15). Specifically, living in a highly urbanized area was found to be associated with schizophrenia [107]. Urban exposure during childhood has been associated with an increased risk of Crohn’s disease and inflammatory bowel disease [108]. Live in a modern house was (compared to traditional house) was found to be a protective factor for clinical malaria [109]. In contrast, living in rural areas has been suggested as a risk factor from Parkinson’s disease [58]. Finally, having pets at home has been suggested to be a protective factor for non-communicable diseases in children and adults (Table 15). Specifically, being exposed to pets in the first year of life was found to reduce the risk of acute lymphoblastic leukemia [110]. For adults, being exposed to a pet was suggested to reduce Crohn’s disease and ulcerative colitis [108].

## 4. Discussion

This umbrella review found 193 associations among 68 environmental exposures and 83 diseases and death causes reported in 101 meta-analyses. The environmental factors found in this review were air pollution, environmental tobacco smoke, heavy metals, chemicals, ambient temperature, noise, radiation, and urban residential surroundings. Among these, we identified 64 environmental exposures defined as risk factors and 4 environmental protective factors. This review offers a comprehensive overview of the latest available evidence on environmental exposures and health outcomes. This, to our knowledge, is the first umbrella review on environmental risk factors and health. We included the most recent meta-analyses that summarize the largest number of individual studies and populations in each research area. We also selected only those meta-estimates that reported statistically significant associations between environmental exposures and health outcomes. In contrast with previous reviews in the area, which only focused on a single exposure or a single health outcome, we provided a general overview of multiples exposures and multiples health outcomes. Furthermore, we focused on observational studies with short and long-term environmental exposures. 

Most of the meta-analyses found were focused on adults (80), 57 included cohorts or case-control studies, and 44 included case-crossover or time series analysis and form all meta-analyses included 79 were published in the last five years. In this review, the largest body of evidence was found in air pollution (91 associations among 14 air pollution definitions and 34 diseases and mortality diagnoses). That could be a reflection of two main factors: a) the relevance of air pollution as the most important environmental risk factor worldwide being one of the top 10 global health risk factors accounting for 4.8 million deaths globally in 2017 [3]; and combined with b) the available research funding, interest, and knowledge to integrate air pollution as an exposure factor in epidemiological studies compared to other pollutants. In terms of air pollution, in this review, particulate matter (PM2.5 and PM10) was the leading pollutant group that reported the largest number of associations (45). Environmental tobacco smoke was the second-largest exposure included in meta-analyses, with 24 associations among 6 exposure definitions. Chemicals (including pesticides) were the third larger group of environmental exposures found among the meta-analyses included, with 19 associations. Four environmental exposures were found to be protective for different health outcomes. These protective factors were residential greenness, modern housing, pet exposure, UV radiation, and recreational sun exposure. Despite the evidence on protective environmental factors, the largest body of evidence found in this review was on environmental risk factors (64 exposure definitions). Most of the meta-analyses included in this review reported observational studies from multiple geographical locations and multiple nations. Although some meta-analyses on specific geographical regions or countries were found during the screening step, we only selected those that included the largest number of observational studies. In all cases, this led to select those meta-analyses that do not restrict by geographical location. 

In terms of the strength of evidence, we only found six associations that were assessed with “high” strength of evidence (defined as those associations that reported precision of the estimate (*p* < 0.001) and consistency of results (I2 < 50%)). The associations with “high” strength of evidence were NO2 and Type 2 diabetes; passive smoking and Type 2 diabetes; 1,3 Butadiene and acute lymphoblastic leukemia; aluminum and dementia; road traffic noise and hypertension; and residential greenness and low birth weight. In all the cases, but 1,3 Butadiene (case-control in children), the associations were reported in cohort studies from adult populations. Based on our definition of the strength of evidence, we consider that those six associations will be the only ones that we do not expect to change in direction (i.e., risk vs. protective factor) or magnitude of the association even if new studies on these topics are published.


This study encountered several limitations that should be considered. As with any systematic review, publication bias was the main limitation. To mitigate this, we focused our research on PubMed publications, where we searched for free text and medical subheadings (MESH) terms. A hand search complemented this effort. One important limitation of this review is the inclusion of a single literature database (Medline via PubMed). We acknowledge that this review will probably only capture the literature published primarily in health journals. Other data sources (i.e., Web of Science) could capture other sectoral journals (i.e., environment). Due to the limited resources and the large scope of this review, we decided to concentrate our resources on “PubMed” because it was considered the primary data source on health evidence. Another limitation we found was the quality of the included studies as most of the examined meta-analyses had a large heterogeneity. This review aimed to include studies focusing on the “environment” defined as the external elements and conditions which surround, influence, and affect the life and development of a human organism or population. While this review considers physical environments such as nature, the built environment, and pollution, it does not consider social environments. This review does not include occupational exposures; water, sanitation, and hygiene (WASH) exposures; behavioral risk factors (e.g., physical activity or diet); or exposure to microorganisms and no-natural disasters. This review selected only those meta-analysis that includes disease prevalence, incidence, and causes of death. The current epidemiological evidence provides a large body of studies (e.g., on biomarkers, metabolic and cardiovascular risk factors, symptoms, sings, hospitalizations, and emergency room visits, among others) that were beyond the scope of this review. We favored health evidence on defined diseases and causes of death that could be more easily translated into public health interventions and practices, although we acknowledge that preclinical and symptomatic health indicators could affect the largest portion of the population. In addition, there are several environmental exposures that were not included in this umbrella review based on the inclusion criteria. For example, large single observational studies were not included in the scope of this review. Furthermore, in the case that several observational studies on the similar exposure and outcome where published this study would be not able to include those type of evidence if those where not combined in a meta-analysis. For those reasons this umbrella review should be considered as a complementary tool to understand the universe of evidence available on environmental health. 

Although this umbrella review found several publications and associations among environmental exposures and health outcomes, we also identified several evidence gaps. Most of the studies focus on identifying environmental risk factors, and only a few studies have been focusing on identifying environmental protective factors. Furthermore, few studies have focused on vulnerable and disadvantaged populations (children, elders, social disadvantaged, ethnic minorities, etc.). Furthermore, most studies do not provide a clear definition of the health outcomes using the international classification of diseases (ICD), nor a comparable exposure definition when the same pollutant is used. In terms of the meta-analysis, we exclude several studies from this review because, in the analyses, cross-sectional studies were mixed with other observational studies (i.e., cohorts). Additionally, several studies did not report heterogeneity values (i.e., I2) or do not provide dose-response functions essential for population risk assessment, health impact assessments and policy translation. We have summarized a list of recommendations for future research in environmental health studies based on these gaps, and we have listed those recommendations in Table 16.

## 5. Conclusions

Environmental exposures are an important health determinant. This umbrella review identified 68 environmental exposures that were associated to 83 health outcomes. This review provides an overview of an evolving area of research and should be used as a complementary tool to understand the connections between the environment and human health. This review also found the need of research prioritization using longitudinal approaches with harmonized exposure and outcome definitions, including vulnerable and susceptible populations in environmental health. The evidence presented by this review should help to design public health interventions and the implementation of a health in all policies approach aiming to improve populational health.

## Figures and Tables

**Figure 1 ijerph-18-00704-f001:**
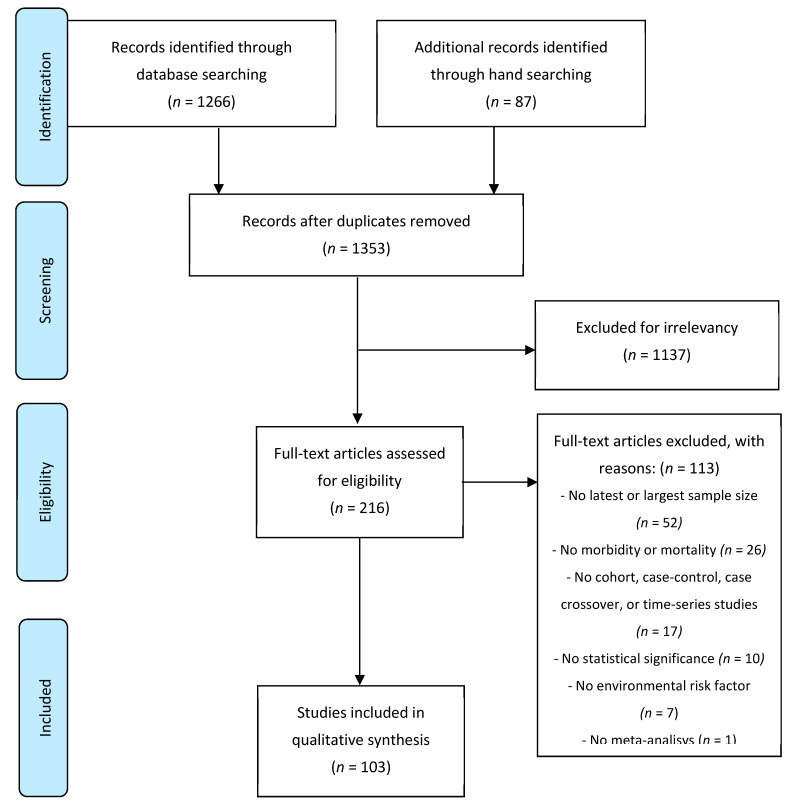
Flow diagram of the study selection.

**Table 1 ijerph-18-00704-t001:** Particulate matter less than 2.5 micrometers of diameter (PM2.5) and long-term health outcomes.

Environmental Risk Factor	Exposure Unit or Comparator	Exposure Temporality	Study Design	Population	Health Outcome	Studies Included	Reference	Year	I2 (%)	*p*-Value	Risk Estimate	Effect Size	LCI	UCI	Strength of Evidence
PM2.5	per 1 mcg/m^3^	Long-term	Cohort	Adults, both sexes	Alzheimer’s disease	3	[9]	2019	86	0	HR	4.82	2.28	7.36	Moderate
	per 10 mcg/m^3^				All-cause mortality	13	[10].	2013	65	0.001	RR	1.06	1.04	1.08	Moderate
					Cardiovascular mortality	17	[11]	2014	98	NR	RR	1.19	1.09	1.31	Low
					Chronic kidney disease	4	[5]	2020	82	0.001	RR	1.10	1.00	1.21	Low
					Chronic Obstructive Pulmonary Disease	4	[8]	2014	NR	NR	IRF	F	F	F	Low
					Dementia	4	[9]	2019	97	0	HR	3.26	1.20	5.31	Moderate
					Depression	5	[12]	2019	0	0.97	OR	1.10	1.02	1.19	Moderate
					Ischemic heart disease mortality	16	[8]	2014	NR	NR	IRF	F	F	F	Low
					Lung cancer mortality	49	[8]	2014	NR	NR	IRF	F	F	F	Low
					Liver cancer mortality	2	[7]	2018	67	NR	RR	1.29	1.06	1.58	Low
					Colorectal cancer mortality	2	[7]	2018	97	NR	RR	1.08	1.00	1.17	Low
					Cancer mortality	19	[7]	2018	97	<0.001	RR	1.17	1.11	1.24	Moderate
					Natural mortality	11	[11]	2014	87	NR	RR	1.05	1.01	1.01	Low
					Respiratory mortality	8	[11]	2014	61	NR	RR	1.05	1.01	1.09	Low
					Stroke	16	[13]	2019	77	0	HR	1.11	1.05	1.17	Moderate
					Stroke mortality	16	[8]	2014	NR	NR	IRF	F	F	F	Low
					Type 2 diabetes	10	[6]	2020	55	0.012	RR	1.11	1.03	1.19	Low
					Parkinson’s disease	8	[14]	2019	86	<0.001	RR	1.06	0.99	1.14	Moderate

LCI: lower confidence intervals; UCI: upper confidence intervals; NR: No reported; HR: hazard rations; RR: relative risk; IRF: integrated response function; F: function; OR: odds ratio.

**Table 2 ijerph-18-00704-t002:** Particulate matter less than 2.5 micrometers of diameter (PM2.5), long-term, and short-term health outcomes.

Environmental Risk Factor	Exposure Unit or Comparator	Exposure Temporality	Study Design	Population	Health Outcome	Studies Included	Reference	Year	I2 (%)	*p*-Value	Risk Estimate	Effect Size	LCI	UCI	Strength of Evidence
PM2.5	per 10 mcg/m^3^	Long-term	Cohort	Children	Asthma	10	[36]	2017	28	0.18	OR	1.03	1.01	1.05	Moderate
					Autism spectrum disorder	3	[20]	2016	0	0.54	OR	2.32	2.15	2.51	Moderate
				Children (<5 years)	Acute low respiratory infections	28	[8]	2014	NR	NR	IRF	F	F	F	Low
				Pregnant women	Small for gestational age	5	[18]	2019	51	NR	OR	1.01	1.00	1.03	Low
					Autistic syndrome disorder	9	[17]	2020	91	<0.001	RR	1.06	1.01	1.11	Moderate
	per 10 mcg/m^3^	Short-term	Case-crossover	Adults, both sexes	Out-of-hospital cardiac arrest	12	[21]	2017	70	NR	RR	1.04	1.01	1.07	Low
			Time-series	Adults, both sexes	Cardiac arrhythmia	17	[22]	2016	78	NR	RR	1.15	1.01	1.03	Low
					Daily cardiovascular mortality	652	[23]	2019	NR	NR	RR	1.36	1.30	1.43	Low
					Daily mortality	652	[23]	2019	NR	NR	RR	1.68	1.59	1.77	Low
					Daily respiratory mortality	652	[23]	2019	NR	NR	RR	1.47	1.35	1.58	Low
				Children (<18 years)	Pneumonia	11	[24]	2017	38	0.08	RR	1.02	1.01	1.03	Moderate

LCI: lower confidence intervals; UCI: upper confidence intervals; NR: No reported; HR: hazard rations; RR: relative risk; IRF: integrated response function; F: function; OR: odds ratio.

**Table 3 ijerph-18-00704-t003:** Particulate matter less than 10 micrometers of diameter (PM10), desert dust, black carbon, long-term and short-term health outcomes.

Environmental Risk Factor	Exposure Unit or Comparator	Exposure Temporality	Study Design	Population	Health Outcome	Studies Included	Reference	Year	I2 (%)	*p*-Value	Risk Estimate	Effect Size	LCI	UCI	Strength of Evidence
PM10	per 2 mcg/m^3^	Long-term	Cohort	Adults, both sexes	Chronic kidney disease	4	[5]	2020	81	0.001	RR	1.16	1.05	1.29	Low
	per 10 mcg/m^3^				Type 2 diabetes	6	[6]	2020	68	0.004	RR	1.12	1.01	1.23	Moderate
					Incidence of coronary events	11	[15]	2014	0	0.81	HR	1.12	1.01	1.25	Moderate
					Lung cancer mortality	9	[7]	2018	93	NR	RR	1.07	1.03	1.11	Low
					Cancer mortality	12	[7]	2018	91	<0.001	RR	1.09	1.04	1.14	Moderate
					Incidence of chronic bronchitis	3	[16]	2015	NR	NR	RR	1.11	1.04	1.18	Low
				Children	Asthma	12	[36]	2017	29	0.16	OR	1.05	1.02	1.08	Moderate
				Pregnant women	Low birth weight	11	[18]	2019	73	NR	OR	1.06	1.02	1.09	Low
					Preterm birth	8	[18]	2019	81	NR	OR	1.05	1.02	1.07	Low
			Case-control	Children	Autism spectrum disorder	6	[20]	2016	2	0.41	OR	1.07	1.06	1.08	Moderate
		Short-term	Case-crossover	Adults, both sexes	Out-of-hospital cardiac arrest	9	[21]	2017	78	NR	RR	1.02	1.01	1.04	Low
			Time-series	Adults, both sexes	Cardiac arrhythmia	12	[22]	2016	79	NR	RR	1.01	1	1.02	Low
					Daily cardiovascular mortality	652	[23]	2019	NR	NR	RR	1.55	1.45	1.66	Low
					Daily mortality	652	[23]	2019	NR	NR	RR	1.44	1.39	1.5	Low
					Daily respiratory mortality	652	[23]	2019	NR	NR	RR	1.74	1.53	1.95	Low
	per 20 mcg/m^3^				Suicide	7	[12]	2019	42	0.15	RR	1.02	1	1.03	Moderate
				Children (<18 years)	Pneumonia	10	[24]	2017	66	0	RR	1.02	1.01	1.02	Moderate
Desert dust	per 10 mcg/m^3^	Short-term	Time-series	Adults, both sexes	Cardiovascular mortality	11	[25]	2016	0	0.77	IR	1.01	1	1.02	Moderate
					Mortality	11	[25]	2016	0	0.75	IR	1.01	1	1.01	Moderate
Black carbon	per 0.5 × 10^−5^ M^−1^	Long-term	Cohort	Children	Asthma	8	[36]	2017	0	0.87	OR	1.08	1.03	1.14	Moderate

LCI: lower confidence intervals; UCI: upper confidence intervals; NR: No reported; HR: hazard rations; RR: relative risk; IRF: integrated response function; F: function; OR: odds ratio.

**Table 4 ijerph-18-00704-t004:** Nitrogen oxides (NOx), nitrogen dioxide (NO2), long-term and short-term health outcomes.

Environmental Risk Factor	Exposure Unit or Comparator	Exposure Temporality	Study Design	Population	Health Outcome	Studies Included	Reference	Year	I2 (%)	*p*-Value	Risk Estimate	Effect Size	LCI	UCI	Strength of Evidence
NO2	per 4 mcg/m^3^	Long-term	Cohort	Adults, both sexes	Autistic syndrome disorder	7	[17]	2020	58	0.007	RR	1.02	1.01	1.04	Low
	per 10 mcg/m^3^				Cancer mortality	16	[7]	2018	95	0.003	RR	1.06	1.02	1.10	Low
					Cardiovascular mortality	18	[11]	2014	98	NR	RR	1.13	1.08	1.18	Low
					Chronic kidney disease	3	[5]	2020	0	0.47	RR	1.11	1.09	1.14	Moderate
					All-cause mortality	12	[11]	2014	89	NR	RR	1.04	1.01	1.06	Low
					Respiratory mortality	9	[11]	2014	0	NR	RR	1.02	1.02	1.03	Moderate
					Type 2 diabetes	6	[26]	2018	46	<0.001	RR	1.11	1.07	1.16	High
					Cancer mortality	16	[7]	2018	95	0.003	RR	1.06	1.02	1.10	Moderate
				Children	Asthma	20	[36]	2017	65	<0.001	OR	1.05	1.02	1.07	Moderate
				Pregnant women	Low birth weight	11	[18]	2019	32	NR	OR	1.02	1.00	1.04	Moderate
					Small for gestational age	5	[18]	2019	87	NR	OR	1.02	1.01	1.03	Low
	per 10 mcg/m^3^	Short-term	Time-series	Adults	Natural mortality	30	[16]	2015	NR	NR	RR	1.002	1.002	1.004	Low
	per 10 ppb		Case-crossover	Adults, both sexes	Out-of-hospital cardiac arrest	11	[21]	2017	66	NR	RR	1.02	1.00	1.03	Low
			Time-series	Adults, both sexes	Cardiac arrhythmia	13	[22]	2016	93	NR	RR	1.04	1.01	1.05	Low
					Conjunctivitis	12	[27]	2019	NR	NR	RR	1.02	1.01	1.04	Low
	per 20 ppb				Depression	7	[28]	2020	65	0.008	RE	1.02	1.00	1.04	Low
				Children (<18 years)	Pneumonia	10	[24]	2017	71	0	RR	1.01	1.00	1.02	Moderate
NOx	per 20 ppb	Long-term	Cohort	Pregnant women	Low birth weight	3	[18]	2019	58	NR	OR	1.03	1.01	1.05	Low
					Preterm birth	5	[18]	2019	88	NR	OR	1.02	1.01	1.03	Low

LCI: lower confidence intervals; UCI: upper confidence intervals NR: No reported; RR: relative risk; OR: odds ratio.

**Table 5 ijerph-18-00704-t005:** Ozone (O3), Sulfur Dioxide (SO2), and Carbon Monoxide (CO), long-term and short-term health outcomes.

Environmental Risk Factor	Exposure Unit or Comparator	Exposure Temporality	Study Design	Population	Health Outcome	Studies Included	Reference	Year	I2 (%)	*p*-Value	Risk Estimate	Effect Size	LCI	UCI	Strength of Evidence
O3	per 5 ppb	Long-term	Cohort	Adults, both sexes	Ischemic heart disease mortality	4	[29]	2016	67	0.02	RR	1.02	1	1.04	Low
	per 10 mcg/m^3^			Pregnant women	Preterm birth	3	[18]	2019	0	NR	OR	1.04	1	1.07	Moderate
	per 10 ppb		Cohort and Case-Control	Adults, both sexes	Parkinson’s disease	5	[14]	2019	0	0.69	RR	1.01	1	1.02	Moderate
		Short-term	Case-crossover	Adults, both sexes	Out-of-hospital cardiac arrest	11	[21]	2017	53	NR	RR	1.02	1.01	1.02	Low
	per 20 ppb		Time-series	Children (<18 years)	Pneumonia	12	[24]	2017	75	0	RR	1.02	1.01	1.03	Moderate
	per 10 mcg/m^3^			Adults	All-cause mortality	32	[16]	2015	NR	NR	RR	1.003	1.001	1.004	Low
					Cardiovascular and respiratory mortality	32	[16]	2015	NR	NR	RR	1.005	1.001	1.009	Low
SO2	per 5 ppb	1st pregnancy trimester	Cohort	Pregnant women	Gestational diabetes mellitus	5	[30]	2020	93	0	OR	1.39	1.01	1.77	Moderate
	per 10 mcg/m^3^	Long-term	Cohort	Pregnant women	Low birth weight	5	[18]	2019	98	NR	OR	1.21	1.08	1.35	Low
	per 10 ppb	Short-term	Time-series	Adults, both sexes	Cardiac arrhythmia	10	[22]	2016	77	NR	RR	1.02	1	1.04	Low
				Children (<18 years)	Pneumonia	8	[24]	2017	48	0.04	RR	1.03	1	1.05	Moderate
CO	per 1 mcg/m^3^	Long-term	Cohort	Pregnant women	Preterm birth	7	[18]	2019	89	NR	OR	1.06	1.04	1.08	Low
	per 1 ppm	Short-term	Case-crossover	Adults, both sexes	Out-of-hospital cardiac arrest	11	[21]	2017	44	NR	RR	1.06	1	1.14	Moderate
			Time-series	Adults, both sexes	Cardiac arrhythmia	12	[22]	2016	90	NR	RR	1.04	1.02	1.06	Low
	per 1000 ppb			Children (<18 years)	Pneumonia	7	[24]	2017	68	0.004	RR	1.01	1	1.02	Low

LCI: lower confidence intervals; UCI: upper confidence intervals NR: No reported; RR: relative risk; OR: odds ratio.

**Table 6 ijerph-18-00704-t006:** Household Air Pollution, indoor air pollution from solid fuel, biomass burning, and long-term health outcomes.

Environmental Risk Factor	Exposure Unit or Comparator	Exposure Temporality	Study Design	Population	Health Outcome	Studies Included	Reference	Year	I2 (%)	*p*-Value	Risk Estimate	Effect Size	LCI	UCI	Strength of Evidence
Household air pollution	Exposed vs. not exposed	Long-term	Case-controls	Adults, both sexes	Cervical cancer	4	[31]	2015	NR	0.45	OR	6.46	3.12	13.36	Low
					Laryngeal cancer	5	[31]	2015	NR	0.49	OR	2.35	1.72	3.21	Low
					Nasopharyngeal cancer	6	[31]	2015	NR	0.06	OR	1.8	1.42	2.29	Low
					Oral cancer	4	[31]	2015	NR	0.93	OR	2.44	1.87	3.19	Low
					Pharyngeal cancer	4	[31]	2015	NR	0.99	OR	3.56	2.22	5.7	Low
Indoor air pollution from solid fuel	Exposed vs. not exposed	Long-term	Cohort	Adults, both sexes	Hypertension	11	[32]	2020	90	0	OR	1.52	1.26	1.85	Moderate
Solid fuel use	Exposed vs. not exposed	Long-term	Cohort	Pregnant women	Low birth weight	12	[33]	2014	28	0.07	OR	1.35	1.23	1.48	Moderate
					Stillbirth	5	[33]	2014	0	0.44	OR	1.29	1.18	1.41	Moderate
					Preterm birth	3	[33]	2014	0	0.39	OR	1.30	1.06	1.59	Moderate
					Intrauterine growth retardation	2	[33]	2014	0	0.89	OR	1.23	1.01	1.49	Moderate
Biomass burning	Exposed vs. not exposed	Long-term	Case-controls	Adults, both sexes	Esophageal squamous cell carcinoma	16	[34]	2019	79	NR	OR	3.02	2.22	4.11	Low
			Cohort and Case-Control	Adults, both sexes	Chronic Obstructive Pulmonary Disease	8	[35]	2017	93	<0.001	OR	2.21	1.3	3.76	Moderate

LCI: lower confidence intervals; UCI: upper confidence intervals NR: No reported; OR: odds ratio.

**Table 7 ijerph-18-00704-t007:** Environmental Tobacco Smoke and long-term health outcomes.

Environmental Risk Factor	Exposure Unit or Comparator	Exposure Temporality	Study Design	Population	Health Outcome	Studies Included	Reference	Year	I2 (%)	*p*-Value	Risk Estimate	Effect Size	LCI	UCI	Strength of Evidence
Environmental tobacco smoke	Exposed vs. not exposed	Long-term	Cohort	Adults, both sexes	Stroke	23	[39]	2017	NR	NR	RR	1.15	1.06	1.24	Low
			Cohort and Case-Control	Women	Lung cancer	41	[40]	2018	NR	<0.05	RR	1.33	1.17	1.51	Low
				Pregnant women	Low birth weight	10	[37]	2008	54	0.009	OR	1.32	1.07	1.63	Moderate
					Small for gestational age	9	[37]	2008	0	0.004	OR	1.21	1.06	1.37	Moderate
Parental smoking	Exposed vs. not exposed	Long-term	Cohort	Children	Childhood obesity	6	[52]	2014	0	NR	RR	1.33	1.23	1.44	Moderate
Paternal smoking	Exposed vs. not exposed	Long-term	Case-controls	Children	Acute myeloid leukemia	17	[54]	2019	0.5	0.003	OR	1.15	1.038	1.275	Moderate
	Exposed vs. not exposed	Long-term	Case-controls	Children	Acute lymphoblastic leukemia	10	[55]	2012	28	0.18	OR	1.15	1.06	1.24	Moderate
Maternal smoking	Exposed vs. not exposed	Long-term	Case-controls	Children	Neuroblastoma	14	[53]	2019	17	NR	OR	1.1	1.0	1.3	Moderate
Passive smoking	Exposed vs. not exposed	Long-term	Case-controls	Adults, both sexes	Lung adenocarcinoma	18	[44]	2014	NR	0.26	OR	1.35	1.23	1.48	Low
					Lung cancer	18	[44]	2014	NR	0.01	OR	1.34	1.24	1.45	Low
					Lung large cell cancer	18	[44]	2014	NR	0.68	OR	1.36	1.04	1.79	Low
					Lung small cell cancer	18	[44]	2014	NR	0.98	OR	1.63	1.31	2.04	Low
					Lung squamous cell carcinoma	18	[44]	2014	NR	0.06	OR	1.36	1.17	1.58	Low
				Pregnant women	Neural tube defects	11	[46]	2018	50	0.02	OR	1.90	1.56	2.31	Low
			Cohort	Adults, both sexes	Cardiovascular disease	38	[42]	2015	66	0	RR	1.23	1.16	1.31	Moderate
					All-cause mortality	11	[42]	2015	69	0	RR	1.18	1.10	1.27	Moderate
					Type 2 diabetes	7	[26]	2018	31	<0.001	RR	1.22	1.10	1.35	High
			Cohort and Case-Control	Women	Breast cancer	51	[41]	2014	75	<0.001	OR	1.62	1.39	1.85	Moderate
					Cervical cancer	14	[43]	2018	64	0	OR	1.70	1.40	2.07	Moderate
			Cohort	Children	Asthma	41	[47]	2020	86	<0.01	OR	1.21	1.15	1.26	Low
					Otitis Media	9	[48]	2014	80	0.04	OR	1.39	1.02	1.89	Low
Prenatal smoke	Exposed vs. not exposed	Long-term	Cohort	Pregnant women	Schizophrenia	7	[49]	2020	71	NR	OR	1.29	1.10	1.51	Low
					Offspring depression	4	[50]	2017	54	0.084	OR	1.20	1.08	1.34	Low
			Cohort and Case-Control		Attention-deficit/hyperactivity disorder	20	[51]	2017	79	0.000	OR	1.60	1.45	1.76	Moderate

LCI: lower confidence intervals; UCI: upper confidence intervals NR: No reported; RR: relative risk; OR: odds ratio.

**Table 8 ijerph-18-00704-t008:** Chemicals and long-term health impacts.

Environmental Risk Factor	Exposure Unit or Comparator	Exposure Temporality	Study Design	Population	Health Outcome	Studies Included	Reference	Year	I2 (%)	*p*-Value	Risk Estimate	Effect Size	LCI	UCI	Strength of Evidence
1,3-Butadiene	High exposed vs. low exposed	Long-term	Case-controls	Children	Acute lymphoblastic leukemia	2	[56]	2019	0	0	RR	1.31	1.11	1.54	High
					All leukemia	2	[56]	2019	28	0.025	RR	1.45	1.08	1.95	Moderate
Bisphenol A	High exposed vs. low exposed	Long-term	Cohort	Adults, both sexes	Diabetes	3	[64]	2015	0	0.55	OR	1.47	1.21	1.80	Moderate
					Obesity	3	[64]	2015	0	0.44	OR	1.67	1.41	1.98	Moderate
Dioxins	High exposed vs. low exposed	Long-term	Cohort	Women	Endometriosis	10	[62]	2019	72	<0.01	OR	1.65	1.14	2.39	Low
Hydrocarbon exposure	Exposed vs. not exposed	Long-term	Cohort and Case-Control	Adults, both sexes	Parkinson’s disease	14	[58]	2016	28	NR	OR	1.36	1.13	1.63	Moderate
Mono (2-ethyl-5-hydroxyhexyl) phthalate	High exposed vs. low exposed	Long-term	Cohort and Case-Control	Women	Endometriosis	6	[65]	2019	44	0.11	OR	1.24	1.00	1.54	Moderate
Organic solvents	Exposed vs. not exposed	Long-term	Cohort and Case-Control	Adults, both sexes	Multiple sclerosis	15	[59]	2015	77	0.06	RR	1.54	1.03	2.29	Low
					Parkinson’s disease	18	[58]	2016	43	NR	OR	1.22	1.01	1.47	Moderate
Polychlorinated biphenyls (PCBs)	High exposed vs. low exposed	Long-term	Cohort	Women	Endometriosis	9	[62]	2019	78	<0.01	OR	1.70	1.20	2.39	Low
	High exposed vs. low exposed	Long-term	Case-controls	Adults, both sexes	Non-Hodgkin Lymphoma	7	[61]	2012	NR	NR	OR	1.43	1.31	1.55	Low
Polychlorinated biphenyls 153	per log2 ng/L	Long-term	Cohort	Children	Bronchitis	7	[63]	2014	NR	0.89	RR	1.06	1.01	1.12	Low
Solvents	Exposed vs. not exposed	Long-term	Cohort and Case-Control	Adults, both sexes	Systemic sclerosis	11	[60]	2018	55	<0.001	OR	2.41	1.73	3.37	Moderate

LCI: lower confidence intervals; UCI: upper confidence intervals NR: No reported; RR: relative risk; OR: odds ratio.

**Table 9 ijerph-18-00704-t009:** Pesticides and health outcomes.

Environmental Risk Factor	Exposure Unit or Comparator	Exposure Temporality	Study Design	Population	Health Outcome	Studies Included	Reference	Year	I2 (%)	*p*-Value	Risk Estimate	Effect Size	LCI	UCI	Strength of Evidence
Pesticides	Exposed vs. not exposed	Long-term	Cohort and Case-Control	Adults, both sexes	Alzheimer’s disease	7	[66]	2016	0	0.885	OR	1.34	1.08	1.67	Moderate
	High exposed vs. low exposed		Cohort and Case-Control	Adults, both sexes	Amyotrophic lateral sclerosis	7	[67]	2016	41	0.16	RR	1.20	1.02	1.41	Moderate
	High exposed vs. low exposed		Case-controls	Children	Brian tumors	18	[68]	2017	0	NR	OR	1.26	1.13	1.14	Moderate
	Exposed vs. not exposed		Case-controls	Adults, both sexes	Myelodysplastic Syndromes	11	[69]	2014	80	0	OR	1.95	1.23	3.09	Moderate
	10 years of exposure vs. no exposure		Cohort	Adults, both sexes	Parkinson’s disease	10	[70]	2018	50	0.032	OR	1.11	1.05	1.18	Low
Residential pesticide exposure	High exposed vs. low exposed	Long-term	Case-controls	Children	Acute lymphoblastic leukemia	8	[74]	2019	NR	NR	OR	1.42	1.13	1.80	Low
					Acute myeloid leukemia	5	[74]	2019	NR	NR	OR	1.90	1.35	2.67	Low
					Childhood leukemia	15	[74]	2019	73	NR	OR	1.57	1.27	1.95	Low

LCI: lower confidence intervals; UCI: upper confidence intervals NR: No reported; RR: relative risk; OR: odds ratio.

**Table 10 ijerph-18-00704-t010:** Pesticides and health outcomes.

Environmental Risk Factor	Exposure Unit or Comparator	Exposure temporality	Study Design	Population	Health outcome	Studies Included	Reference	Year	I2 (%)	*p*-Value	Risk Estimate	Effect Size	LCI	UCI	Strength of Evidence
Chlordane	High exposed vs. low exposed	Long-term	Case-controls	Adults, both sexes	non-Hodgkin lymphoma	8	[73]	2016	17	0.29	OR	1.93	1.51	2.48	Moderate
Diazinon	Exposed vs. not exposed	Long-term	Cohort and Case-Control	Adults, both sexes	non-Hodgkin lymphoma	7	[72]	2017	0	0.668	OR	1.39	1.11	1.73	Moderate
Dichlorodiphenyldichloroethylene (DDE)	High exposed vs. low exposed	Long-term	Case-controls	Adults, both sexes	non-Hodgkin lymphoma	11	[73]	2016	0	0.94	OR	1.38	1.14	1.66	Moderate
	per log2 ng/L	Long-term	Cohort	Children	Bronchitis	7	[63]	2014	NR	0.38	RR	1.05	1.00	1.11	Low
Hexachlorobenzene	High exposed vs. low exposed	Long-term	Case-controls	Adults, both sexes	non-Hodgkin lymphoma	7	[73]	2016	0	0.64	OR	1.54	1.20	1.99	Moderate
Hexachlorocyclohexane	High exposed vs. low exposed	Long-term	Case-controls	Adults, both sexes	non-Hodgkin lymphoma	6	[73]	2016	34	0.17	OR	1.42	1.08	1.87	Moderate
Organochlorine pesticides	High exposed vs. low exposed	Long-term	Case-controls	Adults, both sexes	non-Hodgkin lymphoma	13	[73]	2016	12	0.253	OR	1.40	1.27	1.56	Moderate
			Cohort	Women	Endometriosis	5	[62]	2019	65	0.02	OR	1.97	1.25	3.13	Low
Organophosphate pesticides	Exposed vs. not exposed	Long-term	Cohort and Case-Control	Adults, both sexes	non-Hodgkin lymphoma	10	[72]	2017	41	0.032	OR	1.22	1.04	1.43	Moderate
Paraquat	Exposed vs. not exposed	Long-term	Case-controls	Adults, both sexes	Parkinson’s disease	14	[71]	2019	31	0.126	OR	1.70	1.28	2.25	Moderate

LCI: lower confidence intervals; UCI: upper confidence intervals NR: No reported; OR: odds ratio.

**Table 11 ijerph-18-00704-t011:** Heavy metals, minerals and long-term health outcomes.

Environmental Risk Factor	Exposure Unit or Comparator	Exposure Temporality	Study Design	Population	Health Outcome	Studies Included	Reference	Year	I2 (%)	*p*-Value	Risk Estimate	Effect Size	LCI	UCI	Strength of Evidence
ALUMINUM	Exposed vs. not exposed	Long-term	Cohort	Adults, both sexes	Dementia	8	[45]	2017	6.2	<0.001	OR	1.72	1.33	2.21	High
Asbestos (non-occupational)	Exposed vs. not exposed	Long-term	Cohort and Case-Control	Adults, both sexes	Mesothelioma	27	[75]	2018	99	NR	RR	5.33	2.53	11.23	Low
Cadmium	High exposed vs. low exposed	Long-term	Case-controls	Adults, both sexes	Cancer	3	[76]	2015	0	0.84	RR	1.22	1.13	1.31	Moderate
					Lung Cancer	3	[76]	2015	0	0.41	RR	1.68	1.47	1.92	Moderate
Chromium	High exposed vs. low exposed	Long-term	Case-controls	Adults, both sexes	Schizophrenia	7	[77]	2019	>50	<0.01	SMD	0.32	0.01	0.63	Moderate
Inorganic arsenic	High exposed vs. low exposed	Long-term	Cohort	Adults, both sexes	Type 2 diabetes	3	[78]	2014	39	0.18	RR	1.39	1.06	1.81	Moderate
Lead	High exposed vs. low exposed	Long-term	Cohort and Case-Control	Adults, both sexes	Amyotrophic lateral sclerosis	3	[79]	2020	51	0.01	RR	1.46	1.16	1.83	Low
	Blood levels in mg/L	Long term	Cohort	Children	Mild mental retardation	7	[80]	2005	NR	NR	OR	F	F	F	Low
Silica exposure	Exposed vs. not exposed	Long-term	Cohort and Case-Control	Adults, both sexes	Systemic sclerosis	16	[60]	2018	96	0.002	OR	2.96	1.65	5.29	Low

LCI: lower confidence intervals; UCI: upper confidence intervals NR: No reported; RR: relative risk; OR: odds ratio; SMD: standard median difference; F: function.

**Table 12 ijerph-18-00704-t012:** Ambient temperature and short-term health outcomes.

Environmental Risk Factor	Exposure Unit or Comparator	Exposure Temporality	Study Design	Population	Health Outcome	Studies Included	Reference	Year	I2 (%)	*p*-Value	Risk Estimate	Effect Size	LCI	UCI	Strength of Evidence
Ambient temperature	Maximum suicide temperature 93rd percentile vs. minimum suicide temperature	Short-term	Time-series	Adults, both sexes	Suicide	341	[81]	2019	3.3	NR	RR	1.33	1.30	1.36	Moderate
	Orthopedic procedures during warmer periods of the year	Short-term	Time-series	Adults, both sexes	Post-operative infection	12	[83]	2019	65	0.001	OR	1.16	1.04	1.30	Moderate
	High versus low temperatures	Short-term	Time-series	Pregnant women	Low birth weight	9	[84]	2020	NR	NR	OR	1.07	1.05	1.16	Low
					Stillbirth	2	[84]	2020	27.8	NR	OR	3.39	2.33	4.96	Moderate
Cold	per 1 Celsius degree decrease	Short-term	Time-series	Children <12 years	Asthma	13	[89]	2017	NR	NR	OR	1.07	1.01	1.12	Low
				Elderly	Cardiovascular disease mortality	9	[87]	2016	98	<0.0001	RR	1.01	1.00	1.00	Moderate
					Cerebrovascular mortality	3	[87]	2016	60	0.001	RR	1.01	1.00	1.01	Low
					Intracerebral hemorrhage	2	[87]	2016	0	0.39	RR	1.01	1.01	1.02	Moderate
					Pneumonia	5	[87]	2016	94	<0.0001	RR	1.06	1.01	1.12	Moderate
					Respiratory disease mortality	8	[87]	2016	90	<0.0001	RR	1.02	1.00	1.00	Moderate
	10th and 1st percentile vs. 25th percentile of temperature	Short-term	Time-series	Adults, both sexes	Diabetes mortality	9	[82]	2016	NR	NR	RR	1.11	1.03	1.19	Low
Cold wave	Exposed vs. not exposed	Short-term	Time-series	Adults, both sexes	Cardiovascular mortality	31	[88]	2020	84.3	<0.001	OR	1.54	1.21	1.97	Moderate
Diurnal temperature range	per 10 Celsius degrees	Short-term	Time-series	Adults, both sexes	Mortality	308	[98]	2018	NR	NR	RR	1.03	1.02	1.03	Low
Heat	90th and the 99th percentile vs. 75th percentile of temperature	Short-term	Time-series	Adults, both sexes	Diabetes mortality	9	[82]	2016	NR	NR	RR	1.20	1.12	1.3	Low
	per 1 Celsius degree increase	Short-term	Time-series	Elderly	Acute renal failure	2	[87]	2016	16	0.27	RR	1.01	1.01	1.02	Moderate
					Cardiovascular disease mortality	15	[87]	2016	99	<0.0001	RR	1.03	1.03	1.04	Moderate
					Cerebrovascular mortality	3	[87]	2016	72	0.03	RR	1.01	1.00	1.02	Low
					Diabetes	3	[87]	2016	25	0.26	RR	1.01	1.00	1.01	Moderate
					Ischemic heart disease mortality	3	[87]	2016	81	0.004	RR	1.01	1.00	1.03	Low
					Respiratory disease	11	[87]	2016	82	<0.0001	RR	1.02	1.01	1.04	Moderate
					Respiratory disease mortality	9	[87]	2016	92	<0.0001	RR	1.00	1.00	1.00	Moderate
Heatwave	Exposed vs. not exposed	Short-term	Time-series	Adults, both sexes	Cardiovascular mortality	36	[86]	2019	99	<0.01	RE	1.15	1.09	1.21	Low
					Respiratory mortality	27	[86]	2019	97	<0.01	RE	1.18	1.09	1.28	Low
				Pregnant women	Preterm birth	6	[84]	2020	44.7	0.11	OR	1.16	1.10	1.23	Moderate

LCI: lower confidence intervals; UCI: upper confidence intervals NR: No reported; RR: relative risk; OR: odds ratio.

**Table 13 ijerph-18-00704-t013:** Light, noise, radon, electromagnetic fields, and long-term health outcomes.

Environmental Risk Factor	Exposure Unit or Comparator	Exposure Temporality	Study Design	Population	Health Outcome	Studies Included	Reference	Year	I2 (%)	*p*-Value	Risk Estimate	Effect Size	LCI	UCI	Strength of Evidence
Artificial light exposure at night	High exposed vs. low exposed	Long-term	Case-controls	Women	Breast cancer	6	[91]	2014	1.9	0.4	RR	1.17	1.11	1.24	Moderate
Outdoor light exposure	High exposed vs. low exposed	Long-term	Cohort	Children	Myopia	4	[90]	2019	91	0.02	OR	0.57	0.35	0.92	Low
Ultraviolet radiation	High exposed vs. low exposed	Long-term	Case-controls	Adults, both sexes	Epstein–Barr Virus positive Hodgkin lymphoma	4	[92]	2013	NR	0.10	OR	0.59	0.36	0.96	Low
Recreational sun exposure	High exposed vs. low exposed	Long-term	Case-controls	Adults, both sexes	Non-Hodgkin lymphoma	4	[93]	2008	NR	0.001	OR	0.76	0.63	0.91	Moderate
Extremely low-frequency electromagnetic fields	High exposed vs. low exposed	Long-term	Cohort and Case-Control	Adults, both sexes	Amyotrophic lateral sclerosis	5	[67]	2016	58	0.34	RR	1.30	1.10	1.60	Low
	High vs. low current wiring configuration codes	Long-term	Cohort and Case-Control	Children	Childhood leukemia	6	[99]	1999	NR	NR	OR	1.46	1.05	2.04	Low
Indoor radon	Exposed vs. not exposed	Long-term	Case-controls	Adults, both sexes	Lung cancer	31	[100]	2019	NR	NR	OR	1.14	1.08	1.21	Low
	High exposed vs. low exposed	Long-term	Case-controls	Children	Leukemia	7	[101]	2012	9	0.36	OR	1.37	1.02	1.82	Moderate
Noise	High exposed vs. low exposed	Long-term	Cohort	Adults, both sexes	Diabetes	5	[94]	2018	31	0.18	HR	1.04	1.02	1.07	Moderate
	per 5 dB				Hypertension	5	[95]	2017	51	0.086	RR	1.20	1.09	1.31	Low
Road traffic noise	per 5 dB	Long-term	Cohort	Adults, both sexes	Diabetes	3	[94]	2018	33	0.222	HR	1.07	1.02	1.12	Moderate
	per 10 dB (Lden)				Ischemic heart disease	7	[97]	2018	NR	NR	RR	1.08	1.01	1.15	Low
				Men	Hypertension	2	[96]	2018	0	<0.001	RR	1.62	1.02	1.09	High

LCI: lower confidence intervals; UCI: upper confidence intervals NR: No reported; RR: relative risk; OR: odds ratio; HR: hazard ratio.

**Table 14 ijerph-18-00704-t014:** Greenness, major roads, petrochemical, and long-term health outcomes.

Environmental Risk Factor	Exposure Unit or Comparator	Exposure Temporality	Study Design	Population	Health Outcome	Studies Included	Reference	Year	I2 (%)	*p*-Value	Risk Estimate	Effect Size	LCI	UCI	Strength of Evidence
Petrochemical industrial complexes	Residence >8 km distance from petrochemical industrial complexes	Long-term	Cohort and Case-Control	Adults, both sexes	Acute myeloid leukemia	7	[106]	2020	50	0.01	RR	1.61	1.12	2.31	Low
					Chronic lymphocytic leukemia	7	[106]	2020	92	0.048	RR	1.85	1.01	6.42	Low
					Leukemia	13	[106]	2020	73	0.001	RR	1.36	1.14	1.62	Low
Proximity to major roadways	Exposed vs. not exposed	Long-term	Cohort	Adults, both sexes	Type 2 diabetes	6	[104]	2017	36	0.025	RR	1.13	1.02	1.27	Moderate
Residential traffic exposure	High exposed vs. low exposed	Long-term	Case-controls	Children	Childhood leukemia	7	[105]	2014	57	0.02	OR	1.39	1.03	1.88	Low
Residential greenness	per 0.1 NDVI within 300 m buffer from residence	Long-term	Cohort	Adults, both sexes	All-cause mortality	9	[102]	2019	95	<0.001	HR	0.96	0.94	0.97	Low
					Low birth weight	10	[103]	2020	41	<0.001	RR	0.98	0.97	0.99	High
	per 0.1 NDVI within 500 m buffer from residence				Small for gestational age	13	[103]	2020	59	0.037	RR	0.99	0.98	1.00	Low

LCI: lower confidence intervals; UCI: upper confidence intervals NR: No reported; RR: relative risk; OR: odds ratio; HR: hazard ratio.

**Table 15 ijerph-18-00704-t015:** Urbanization, pets, and long-term health impacts.

Environmental Risk Factor	Exposure Unit or Comparator	Exposure Temporality	Study Design	Population	Health Outcome	Studies Included	Reference	Year	I2 (%)	*p*-Value	Risk Estimate	Effect Size	LCI	UCI	Strength of Evidence
Rural living	Exposed vs. not exposed	Long-term	Cohort and Case-Control	Adults, both sexes	Parkinson’s disease	31	[58]	2016	78	NR	OR	1.32	1.18	1.48	Low
Urban exposure during childhood	Rural exposure during childhood	Long-term	Case-controls	Adults, both sexes	Crohn’s disease	12	[108]	2019	71	0	OR	1.45	1.14	1.85	Moderate
			Cohort and Case-Control	Adults, both sexes	Inflammatory bowel disease	4	[108]	2019	71	0	OR	1.35	1.15	1.58	Moderate
Urbanicity	Highest vs. lowest category	Long-term	Cohort	Adults, both sexes	Schizophrenia	8	[107]	2018	99	0	OR	2.39	1.62	3.51	Moderate
Modern housing	Exposed vs. not exposed	Long-term	Cohort	Adults, both sexes	Clinical malaria	3	[109]	2015	67	0.05	OR	0.55	0.36	0.84	Low
Pet in the first year of life	Exposed vs. not exposed	Long-term	Case-controls	Children	Acute lymphoblastic leukemia	12	[110]	2018	39	0.08	OR	0.91	0.82	1.00	Low
Pet	Exposed vs. not exposed	Long-term	Cohort and Case-Control	Adults, both sexes	Crohn’s disease	14	[108]	2019	67	0	OR	0.77	0.59	0.94	Moderate

LCI: lower confidence intervals; UCI: upper confidence intervals NR: No reported; RR: relative risk; OR: odds ratio; HR: hazard ratio.

**Table 16 ijerph-18-00704-t016:** Recommendations on observational studies and meta-analyses in environmental health.

Recommendations
Observational studies:< - Increase studies on protective environmental risk factors
- Increase studies on vulnerable and disadvantaged populations
- Provide international classification of diseases (ICD) codes as part of the definitions for health outcomes
- Use comparable exposure definitions for environmental risk factors - Support longitudinal study designs
Meta-analyses - Avoid combining cross-sectional studies with cohort or case-control studies in the meta-estimates
- Provide heterogeneity values (i.e., I2)
- Provide dose-response functions to support populational risk assessment, quantitative health impact assessments, and policy translation

## Data Availability

No new data were created or analyzed in this study. Data sharing is not applicable to this article.

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
