# Peer review of "Environmental Risk Factors and Health: An Umbrella Review of Meta-Analyses"

_ijerph, 2021, doi:10.3390/ijerph18020704_

Round 1
Reviewer 1 Report
The authors of this paper have carried out an umbrella review of meta-analyses to look at the associations between environmental risk factors and health. Not surprisingly a large number of associations were ill-health were reported. A key item in such analyses is the search strategy and the inclusion/exclusion criteria for the selected studies. Unfortunately, there appears to be some confusion at this step of the analysis. The abstract initially states (p30) that the review was conducted on meta-analyses of longitudinal studies (lines 29/30) but later the abstract indicates that the met-analyses included studies with a case-control design. Case-control studies are not longitudinal studies (unless there are nested within a cohort) and hence these meta-analyses should be excluded from this review. Looking at the reviews there are a limited number of such meta-analyses and by excluding them it will strengthen the conclusions of this paper.
The authors report that this review 193 associations among 68 environmental exposures and 83 diseases and death causes reported in 101 meta-analyses (Abstract lines 41/42). It is not clear to me how independent some of these analyses really are. For example, in Table 7 all the 5 health outcomes associated with passive smoking are from one meta-analysis of 18 studies. Are these truly independent meta-analyses? I doubt it. The authors should consider how to report multiple outcomes from the same study.
The authors clearly describe how they assess the strength of evidence in section 2.3 but then don’t seem to use this in the description of the results. This is inappropriate as it equates studies whose strength of evidence is low with those whose strength of evidence is high ie the focus really should be on those studies where the strength of evidence is high.
The authors in the abstract state that the “largest body of evidence was found for air pollution..” . Whilst it may be the case that air pollution is the major environmental cause of ill-health , this could simply reflect the ability to get funding for air pollution studies rather than some other environmental exposure. This needs to be reflected in the discussion.
Table 16 provides some recommendations on longitudinal studies and meta-analyses in environmental health but there is little discussion on these recommendations in the actual discussion. How did the authors come to these conclusions?
There are in addition some minor issues that need to resolved:
- In all Tables the references must be cited by number and not by name so that they can be easily identified in the list of references at the end (which is in numerical rather than alphabetical order). At present it is not easy to identify any specific reference
- The authors need to revisited their way of categorising the exposures in these Tables : for example Table 11 (“Heavy metals and long term health outcomes”) contains both asbestos and silica but neither of them are heavy metals .
- What is “sulfate dioxide”? The commonly used name is sulfur dioxide.
Author Response
Manuscript ID: ijerph-1007397
Title: Environmental risk factors and health: an umbrella review of meta-analyses
Response to Reviewer 1 Comments
We sincerely appreciate your invaluable comments to improve this manuscript. We have revised it and addressed the comments you made in the revised file. Please see the paragraphs below for our reply to each comment. Many thanks!
Comment: The authors of this paper have carried out an umbrella review of meta-analyses to look at the associations between environmental risk factors and health. Not surprisingly a large number of associations were ill-health were reported. A key item in such analyses is the search strategy and the inclusion/exclusion criteria for the selected studies. Unfortunately, there appears to be some confusion at this step of the analysis. The abstract initially states (p30) that the review was conducted on meta-analyses of longitudinal studies (lines 29/30) but later the abstract indicates that the met-analyses included studies with a case-control design. Case-control studies are not longitudinal studies (unless there are nested within a cohort) and hence these meta-analyses should be excluded from this review. Looking at the reviews there are a limited number of such meta-analyses and by excluding them it will strengthen the conclusions of this paper.
Answer: Thanks for your comment. We understand the point raised by the reviewer. We have updated the manuscript to avoid any misunderstanding of the inclusion criteria. We have decided to keep case-control studies as part of this umbrella review. Several meta-analyses include case-control studies (separated or combined with cohort studies). This umbrella review's original aim was to include case-control studies. We prefer not to remove those studies from the review, mainly because there are many health outcomes (i.e., cancer) that will be notably reduced in the review if case-control studies are removed. We have corrected the text and eliminated the term "longitudinal" in the abstract, introduction, methods, results, and discussion.
Comment: The authors report that this review 193 associations among 68 environmental exposures and 83 diseases and death causes reported in 101 meta-analyses (Abstract lines 41/42). It is not clear to me how independent some of these analyses really are. For example, in Table 7 all the 5 health outcomes associated with passive smoking are from one meta-analysis of 18 studies. Are these truly independent meta-analyses? I doubt it. The authors should consider how to report multiple outcomes from the same study.
Answer: Thanks for your comment. We have updated the description to clarify that the number of publications, exposures, and outcomes, instead of meta-analysis, to avoid any misleading.
Now the text says: “This umbrella review found 197 associations among 69 environmental exposures and 83 diseases and death causes reported in 103 publications.”
Comment: The authors clearly describe how they assess the strength of evidence in section 2.3 but then don’t seem to use this in the description of the results. This is inappropriate as it equates studies whose strength of evidence is low with those whose strength of evidence is high ie the focus really should be on those studies where the strength of evidence is high.
Answer: Thanks for your comment. We have added a description of the strength of evidence in the discussion section. Now we are highlighting those associations with "high" strength of evidence.
Now the discussion says: “In terms of the strength of evidence, we only found six associations that were assessed with “high” strength of evidence (defined as those associations that reported precision of the estimate (P < 0.001) and consistency of results (I2 < 50%)). The associations with “high” strength of evidence were NO2 and Type 2 diabetes; passive smoking and Type 2 diabetes; 1,3 Butadiene and acute lymphoblastic leukemia; aluminum and dementia; road traffic noise and hypertension; and residential greenness and low birth weight. In all the cases, but 1,3 Butadiene (case-control in children), the associations were reported in cohort studies from adult populations. Based on our definition of the strength of evidence, we consider that those six associations will be the only ones that we do not expect to change in direction (i.e., risk vs. protective factor) or magnitude of the association even if new studies on these topics are published.”
Comment: The authors in the abstract state that the “largest body of evidence was found for air pollution..” . Whilst it may be the case that air pollution is the major environmental cause of ill-health, this could simply reflect the ability to get funding for air pollution studies rather than some other environmental exposure. This needs to be reflected in the discussion.
Answer: Thanks for your comment. We agree with the reviewer on this point, and we have added a sentence reflecting this issue related to air pollution research.
Now the discussion says: “That could be a reflection of two main factors: a) the relevance of air pollution as the most important environmental risk factor worldwide being one of the top 10 global health risk factors accounting for 4.8 million deaths globally in 2017; and combined with b) the available research funding, interest, and knowledge to integrate air pollution as an exposure factor in epidemiological studies compared to other pollutants.”
Comment: Table 16 provides some recommendations on longitudinal studies and meta-analyses in environmental health but there is little discussion on these recommendations in the actual discussion. How did the authors come to these conclusions?
Answer: Thanks for your comment. We have added a paragraph clarifying the origin of those recommendations.
Now the discussion says: “Although this umbrella review found several publications and associations among environmental exposures and health outcomes, we also identified several evidence gaps. Most of the studies focus on identifying environmental risk factors, and only a few studies have been focusing on identifying environmental protective factors. Also, few studies have focused on vulnerable and disadvantaged populations (children, elders, social disadvantaged, ethnic minorities, etc.). Also, most studies do not provide a clear definition of the health outcomes using the international classification of diseases (ICD), nor a comparable exposure definition when the same pollutant is used. In terms of the meta-analysis, we exclude several studies from this review because, in the analyses, cross-sectional studies were mixed with other observational studies (i.e., cohorts). Additionally, several studies did not report heterogeneity values (i.e., I2) or do not provide dose-response functions essential for population risk assessment, health impact assessments, and policy translation. We have summarized a list of recommendations for future research in environmental health studies based on these gaps, and we have listed those recommendations in table 16.”
Comment: In all Tables the references must be cited by number and not by name so that they can be easily identified in the list of references at the end (which is in numerical rather than alphabetical order). At present it is not easy to identify any specific reference
Answer: Thanks for your comment. We have updated all the reference in the tables to reflect the number of reference.
Comment: The authors need to revisited their way of categorising the exposures in these Tables : for example Table 11 (“Heavy metals and long term health outcomes”) contains both asbestos and silica but neither of them are heavy metals .
Answer: Thanks for your comment. We have updated the title in table 11 to properly reflect the content of the table.
Now the table says: “Table 11. Heavy metals, minerals and long-term health outcomes.”
Comment: What is “sulfate dioxide”? The commonly used name is sulfur dioxide.
Answer: Thanks for your comment. We have corrected this error in the manuscript.

Reviewer 2 Report
The authors of the work analyze environmental threats: air pollution, environmental tobacco smoke, heavy metals, chemicals, ambient temperature, noise, radiation, and urban residential surroundings. Evaluated the associations between environmental risk factors and health. An umbrella review was conducted on meta-analyses of longitudinal studies that evaluated the associations between environmental risk factors and health. This umbrella review found 193 associations among 68 environmental exposures and 83 diseases and death causes reported in 101 meta-analyses.
I have objections about the description of the research methodology. It is difficult to find the studied factors in the chapter "Methodology". Some are in the "Abstract" chapter and others in "Results". Besides, I find the work very interesting and I have no other objections.
Author Response
Manuscript ID: ijerph-1007397
Title: Environmental risk factors and health: an umbrella review of meta-analyses
Response to Reviewer 2 Comments
The authors of the work analyze environmental threats: air pollution, environmental tobacco smoke, heavy metals, chemicals, ambient temperature, noise, radiation, and urban residential surroundings. Evaluated the associations between environmental risk factors and health. An umbrella review was conducted on meta-analyses of longitudinal studies that evaluated the associations between environmental risk factors and health. This umbrella review found 193 associations among 68 environmental exposures and 83 diseases and death causes reported in 101 meta-analyses.
Comment: I have objections about the description of the research methodology. It is difficult to find the studied factors in the chapter "Methodology". Some are in the "Abstract" chapter and others in "Results". Besides, I find the work very interesting and I have no other objections.
Answer: Thanks for your comment. We understand and value your comment. The studied factors reported in the abstract and results are a description of the findings. The literature review was open to identify any exposure factors reported in the literature, and the search strategy and inclusion criteria reflect this intention. That is the reason that we do not report a closed list of exposure factors in the methods. In the method section, we do describe the search strategy used, where the exposure definition was described as "(Environment* OR "Environment" [Mesh] OR Environmental pollution OR "Environmental Pollution" [Mesh] OR Environmental exposures OR "Environmental Exposure" [Mesh] OR Environment Design OR "Environment Design" [Mesh] OR Built Environment OR "Built Environment" [Mesh] OR Environmental Medicine OR "Environmental Medicine" [Mesh] OR Decontamination OR "Decontamination" [Mesh])." In addition, in the methods section, we also included the definitions of "risk factor" and "environment" used in this review (Risk factors were defined as any attribute, characteristic, or exposure of an individual that increases the likelihood of developing a disease or death. The environment was defined as the external elements and conditions that surround, influence, and affect a human organism or population's life and development. The environment definition included the physical environment such as nature, built environment, or pollution, but not the social environment. We excluded occupational exposures, microorganisms, water, sanitation, and hygiene (WASH), behavioral risk factors, and no-natural disasters.). We hope this information clarifies the reviewer's comment.

Reviewer 3 Report
The paper includes an umbrella review of various papers that have conducted a meta-analyses of associations between environmental risk factors and health. This is an important piece of work, but could be strengthened in a number of ways. Firstly, the search has been limited to only one database. The search could have been broadened to include data bases such as Web of Science which usually captures journals that publish articles from an environmental perspective as well. It will be helpful if data extraction included the geographic focus of the studies included in each meta-analyses (e.g. global, Asia Pacific, developing countries etc) and also include that in the discussion section. There are studies missing in the review that may meet the selection criteria-, a couple of examples below: Chen, J. J., M. G. Zhou, J. Yang, P. Yin, B. G. Wang, C. Q. Ou and Q. Y. Liu (2020). "The modifying effects of heat and cold wave characteristics on cardiovascular mortality in 31 major Chinese cities." Environmental Research Letters 15(10). Chersich, M. F., M. D. Pham, A. Areal, M. M. Haghighi, A. Manyuchi, C. P. Swift, B. Wernecke, M. Robinson, R. Hetem, M. Boeckmann, S. Hajat and G. Climate Change Heat-Hlth Study (2020). "Associations between high temperatures in pregnancy and risk of preterm birth, low birth weight, and stillbirths: systematic review and meta-analysis." Bmj-British Medical Journal 371.Author Response
Manuscript ID: ijerph-1007397
Title: Environmental risk factors and health: an umbrella review of meta-analyses
Response to Reviewer 3 Comments
The paper includes an umbrella review of various papers that have conducted a meta-analyses of associations between environmental risk factors and health. This is an important piece of work, but could be strengthened in a number of ways.
Comment: Firstly, the search has been limited to only one database. The search could have been broadened to include data bases such as Web of Science which usually captures journals that publish articles from an environmental perspective as well.
Answer: Thanks for your comment. We agree with the reviewer that adding more data sources will enrich the review. But because this was an extensive review in terms of scope (environmental risk factors and health outcomes), results (103 studies included), and being Medline via PubMed, the most extensive database in health sciences, we decided to focus our limited resources on this primary database. We acknowledge the limitations of this decision and lack of resources, and we have included this limitation in the discussion section.
Now the discussion says: “One important limitation of this review is the inclusion of a single literature database (Medline via PubMed). We acknowledge that this review will probably only capture the literature published primarily in health journals. Other data sources (i.e., Web of Science) could capture other sectoral journals (i.e., environment). Due to the limited resources and the large scope of this review, we decided to concentrate our resources on "PubMed" because it was considered the primary data source on health evidence.”
Comment: It will be helpful if data extraction included the geographic focus of the studies included in each meta-analyses (e.g. global, Asia Pacific, developing countries etc) and also include that in the discussion section.
Answer: Thanks for your comment. We also agree with the reviewer that the geographical description could be valuable information. In this case, most of the meta-analyses included multinational studies. Although we found some meta-analyses on specific geographical regions or countries during the screening step, we only selected those that included the largest number of observational studies. In all cases, this led to select those meta-analyses that do not restrict by geographical location. We have added a sentence in the discussion to clarify this point.
Now the discussion says: “Most of the meta-analyses included in this review reported observational studies from multiple geographical locations and multiple nations. Although some meta-analyses on specific geographical regions or countries were found during the screening step, we only selected those that included the largest number of observational studies. In all cases, this led to select those meta-analyses that do not restrict by geographical location.”
Comment: There are studies missing in the review that may meet the selection criteria-, a couple of examples below: Chen, J. J., M. G. Zhou, J. Yang, P. Yin, B. G. Wang, C. Q. Ou and Q. Y. Liu (2020). "The modifying effects of heat and cold wave characteristics on cardiovascular mortality in 31 major Chinese cities." Environmental Research Letters 15(10). Chersich, M. F., M. D. Pham, A. Areal, M. M. Haghighi, A. Manyuchi, C. P. Swift, B. Wernecke, M. Robinson, R. Hetem, M. Boeckmann, S. Hajat and G. Climate Change Heat-Hlth Study (2020). "Associations between high temperatures in pregnancy and risk of preterm birth, low birth weight, and stillbirths: systematic review and meta-analysis." Bmj-British Medical Journal 371.
Answer: Thanks for your comment. We have reviewed and added these two references to the umbrella review.

Round 2
Reviewer 1 Report
The authors have addressed the comments I raised first time around.